# Cycle-Consistent Learning for Joint Layout-to-Image Generation and Object Detection

## Abstract

In this paper, we propose a **generation-detection cycle consistent** (GDCC) learning framework that jointly optimizes both layout-to-image (L2I) generation and object detection (OD) tasks in an end-to-end manner. The key of GDCC lies in the inherent duality between the two tasks, where L2I takes all object boxes and labels as input conditions to generate images, and OD maps images back to these layout conditions. Specifically, in GDCC, L2I generation is guided by a layout translation cycle loss, ensuring that the layouts used to generate images align with those predicted from the synthesized images. Similarly, OD benefits from an image translation cycle loss, which enforces consistency between the synthesized images fed into the detector and those generated from predicted layouts. While current L2I and OD tasks benefit from large-scale annotated layout-image pairs, our GDCC enables more efficient use of unpaired layout data, thereby further enhancing data efficiency. It is worth noting that our GDCC framework is computationally efficient thanks to the perturbative single-step sampling strategy and a priority timestep re-sampling strategy during training, while maintaining the same inference cost as the original L2I and OD models. Extensive experiments demonstrate that GDCC significantly improves the controllability of diffusion models and the accuracy of object detectors. Our code will be released.

## 1 Introduction

Recent advancements in both layout-to-image (L2I) generation [36] and object detection (OD) [20] tasks have achieved remarkable success, largely driven by the availability of large-scale annotated datasets. Specifically, L2I generation methods incorporate image-based [36; 75; 33] or prompt-based [6; 73] conditional controls into text-to-image (T2I) diffusion models [51] to achieve more precise control over the instance placement during image synthesis. These methods train diffusion models to generate realistic images from structured layouts, which include bounding boxes and object class labels that define the spatial positioning and types of objects in the scene. On the other hand, OD takes an image as input and identifies the objects within it by predicting their bounding boxes and class labels. Current advancements have led to significant improvements in the precision of instance placement for L2I generation and the prediction accuracy of OD models.

Although both L2I generation and OD have been extensively studied, few have noticed the strong correlation between these two tasks, *i.e.*, they can be viewed as inverse tasks of each other, where L2I maps layouts to images and OD maps images to layouts. This natural duality between these two tasks has largely been overlooked in previous research. Our key finding is that such duality can be effectively leveraged to improve the performance of both tasks. Specifically, if we map an image to its corresponding layout using an OD model, and then map that layout back to an image using an L2I model, we should ideally recover the original image. Similarly, mapping a layout to an image and then mapping that image back should yield the original layout. This *cycle consistency* not only enforces tighter alignment between the two tasks but also provides a natural regularization that enhances the learning processes of both tasks. Moreover, the cycle consistency allows for the use of unpaired data, opening up new possibilities for improving data efficiency.

Based on the above insight, in this paper, we are the first to propose a **generation-detection cycle consistent** (GDCC) learning framework that jointly optimizes L2I generation and OD in an end-to-end manner. In GDCC, consistency is maintained in two directions through two key components:

Figure 1: **Overall comparison. (a)** Some works such as [33] use a pre-trained discriminative reward model $\mathcal{R}$ to fine-tune the L2I generator $\mathcal{G}$. **(b)** Some [6; 67] show that the synthesized images provided by a pre-trained $\mathcal{G}$ can improve the performance of the object detector $\mathcal{D}$. **(c)** GDCC enables mutual enhancement between $\mathcal{G}$ and $\mathcal{D}$ through cycle-consistent learning. See §1 for details.

(i) the **layout translation cycle loss**, which ensures consistency between the original layouts used to generate images and those predicted from the synthesized images, and (ii) the **image translation cycle loss**, which enforces consistency between the synthesized images and those reconstructed from the layouts predicted by the detector. These two losses guide the learning process in a cycle-consistent manner, ensuring tight alignment between the tasks during training and fostering mutual enhancement, which leads to more controllable diffusion models and more accurate object detectors.

Our GDCC framework offers several key advantages. First, GDCC enables mutual enhancement between L2I generation and OD, setting it apart from earlier approaches that focus on using one task to improve the other [33; 6; 67]. Such mutual enhancement results in more powerful L2I or OD models, as opposed to relying on pre-trained ones that are not fully optimized for improving the other task and may introduce errors during the training. Second, GDCC demonstrates superior data efficiency by effectively utilizing unpaired layout data, a capability not achieved by previous methods. Third, GDCC is computationally efficient in both training and inference. Our training process is accelerated by a perturbative single-step sampling strategy and a priority timestep re-sampling strategy, and our inference cost remains unchanged because the original network architectures of the L2I and OD models are preserved.

The key contributions of this paper are as follows:

- We are the first to identify the duality between L2I generation and OD, an insight that has previously been overlooked in the literature.

- Inspired by the task duality, we propose a **generation-detection cycle consistent** (GDCC) framework that jointly optimizes both tasks in an end-to-end manner and enables mutual enhancement between them.

- Our GDCC demonstrates both data and computational efficiency by allowing for the use of unpaired data and incorporating a perturbative single-step sampling strategy along with a priority timestep re-sampling strategy to accelerate training.

Extensive experimental results confirm that GDCC establishes new benchmarks in both L2I generation and OD. For L2I generation, it achieves up to a 2.1% FID improvement over baseline L2I methods, and shows a 2.1% increase in YOLO score, indicating superior alignment between generated images and conditional layouts. For OD, GDCC achieves up to a 0.9% point gain in AP, further validating the mutual enhancement between L2I generation and OD tasks. These results confirm the effectiveness of our cycle-consistent framework in improving the controllability of diffusion models for image synthesis and the accuracy of object detectors.

## 2 RELATED WORK

**Diffusion Models.** Diffusion probabilistic models, first introduced in [56], have witnessed significant advancements both theoretically [13; 24; 31] and methodologically [25; 57; 58] in recent years. Latent Diffusion Model [51] further reduces computational costs by applying the diffusion process in the latent feature space rather than the pixel space. Due to their exceptional sample quality, diffusion models have set new standards across various benchmarks [11; 63; 72], including image editing [2; 29; 40; 44; 22], image-to-image transformation [53; 62; 32], and text-to-image (T2I) generation [45; 46; 49; 48; 51; 54; 16]. Recent layout-to-image (L2I) studies seek to achieve more precise control over instance placement by extending pre-trained T2I models with layout conditions such as bounding boxes and object labels. Early approaches [76; 36; 27; 74; 59; 60] relied

on a closed-set vocabulary from training labels (*e.g.*, COCO [3]) without using text prompts. With the emergence of image-text models such as CLIP [47], open-vocabulary methods became feasible [77; 9; 73; 6; 67; 10; 69; 7]. These methods incorporate layout information as text embeddings into pre-trained T2I diffusion models [51] to achieve more precise control over instance positioning.

In this paper, we boost L2I generation performance from a new perspective by proposing a cycle-consistent learning framework to achieve mutual benefits with OD, which naturally performs the inverse mapping of L2I from images to layouts. Our framework is computationally efficient thanks to the perturbative single-step sampling strategy and a priority timestep re-sampling strategy during training, while maintaining the same inference cost as the original L2I and OD models.

**L2I Generation and OD.** Several works have involved both L2I and OD tasks, but primarily use one to enhance the other. For example, ControlNet++ [33] uses pre-trained discriminative reward models to fine-tune controllable diffusion models. However, these reward models are constrained by their original training data and struggle to adapt to the styles of synthesized images, which hinders their ability to provide more accurate feedback signals for training L2I models. On the other hand, GeoDiffusion [6] demonstrates that OD can benefit from high-quality synthesized data generated by L2I models. DetDiffusion [67] further exploits the synergy between L2I and perceptive models (*e.g.*, semantic segmentation models) to enhance generation controllability, and show that the synthesized images can improve the performance in downstream tasks such as OD. Despite these advances, the potential of tuning L2I models to generate samples specifically designed to boost OD performance remains underexplored.

This paper, for the first time, fully recognizes the duality between L2I and OD tasks and proposes a unified framework GDCC that enables *mutual enhancement* between the two tasks. Furthermore, in addition to leveraging large-scale paired layout-image data, our framework can effectively utilize unpaired layout data, resulting in superior data efficiency.

**Cycle-Consistent Learning.** Cycle-consistent learning is a technique that leverages cyclic transformations to regularize the training process, ensuring that the data or tasks remain aligned when converted back and forth between representations. It can be applied within a single task through sample cycling, such as object tracking [65; 43; 68], temporal representation learning [14] and image generation [78; 30; 71; 37; 8; 33]. It has also been shown to improve model performance across related tasks such as question answering *v.s.* question generation [61; 55; 34], captioning *v.s.* grounding [18; 66], vision-language navigation *v.s.* instruction generation [64], *etc*.

In this paper, we explore the uncharted potential of cycle-consistent learning between L2I generation and OD tasks, wherein the correlation and inherent duality have long been overlooked. These two tasks are seamlessly integrated into an end-to-end cycle-consistent learning framework, where their symmetrical structures provide informative feedback signals that enhance each other. Moreover, our framework allows for the usage of unpaired layout data, leading to superior data efficiency.

## 3 METHODOLOGY

In §3.1, we first introduce the preliminaries of diffusion-based L2I generation and OD. In §3.2.1, we then explore the inherent duality between these two tasks and show how our GDCC leverages cycle consistency to achieve mutual improvement. Finally, we present GDCC in both paired (§3.2.2) and unpaired (§3.2.3) data settings.

### 3.1 PRELIMINARY

**Diffusion-based L2I Generation.** Diffusion models (DMs) [11; 13; 25], functioning by progressively transforming an initial random noise distribution into a coherent image, have arisen as renowned T2I generation methods. DMs define a $T$-step Markovian diffusion forward process to add Gaussian noise $\epsilon$ into input image $x_0$:

$$x_t = \sqrt{\bar{\alpha}_t}x_0 + \sqrt{1 - \bar{\alpha}_t}\epsilon, \quad \epsilon \sim \mathcal{N}(\mathbf{0}, I), \tag{1}$$

where $x_t$ is the perturbed image, $t$ is the timestep, $\bar{\alpha}_t = \prod_{s=0}^{t} \alpha_s$, and $\alpha_t = 1 - \beta_t$ is a differentiable function of $t$ determined by the denoising sampler.

Diffusion-based L2I generation introduces additional control over DMs by incorporating layout conditions. Given a text prompt $\boldsymbol{y}$ and a layout condition $\boldsymbol{l}$, the training loss can be formulated as:

$$\mathcal{L}_{\mathrm{dm}} = \mathbb{E}_{t,\boldsymbol{x}_0,\boldsymbol{y},\boldsymbol{l},\boldsymbol{\epsilon}\sim\mathcal{N}(0,1)} \left\| \boldsymbol{\epsilon} - \boldsymbol{\epsilon}_\theta\big(t, \boldsymbol{x}_t, \boldsymbol{y}, \boldsymbol{l}\big) \right\|_2^2, \tag{2}$$

where $\boldsymbol{\epsilon}_\theta$ is the noise predictor realized as a U-Net [52].

During the sampling stage of L2I generation, the denoising process progressively eliminates the noise estimated by the diffusion model from a randomly sampled noise to predict the final image. Given a random noise $\boldsymbol{\epsilon}$, conditional text $\boldsymbol{y}$, and layout $\boldsymbol{l}$, the sampling process can be simplified to:

$$\boldsymbol{x}^{\mathrm{syn}} = \mathcal{G}^T\big(t, \boldsymbol{\epsilon}, \boldsymbol{y}, \boldsymbol{l}\big), \quad \boldsymbol{\epsilon} \sim \mathcal{N}(\boldsymbol{0}, I), \tag{3}$$

where $\boldsymbol{x}^{\mathrm{syn}} \in \mathbb{R}^{H\times W\times 3}$ represents the synthesized image, and $\mathcal{G}^T$ denotes an L2I generator that performs $T$ denoising steps. The layout $\boldsymbol{l} = \{(\boldsymbol{b}_n, c_n)\}_{n=1}^N \in \mathbb{R}^{N\times 5}$ consists of $N$ bounding boxes, where each bounding box $\boldsymbol{b}_n = [x_{n,1}, y_{n,1}, x_{n,2}, y_{n,2}]$ defines the spatial location of object $n$, and $c_n \in \mathcal{C}$ denotes its corresponding semantic class.

**Object Detection.** This task aims to train a detector $\mathcal{D}(\cdot)$ to identify and localize objects within an image by predicting bounding boxes and their corresponding class labels:

$$\boldsymbol{l} = \mathcal{D}\big(\boldsymbol{x}\big), \tag{4}$$

where $\boldsymbol{x} \in \mathbb{R}^{H\times W\times 3}$ denotes the input image, and $\boldsymbol{l} = \{(\boldsymbol{b}_n, c_n)\}_{n=1}^N \in \mathbb{R}^{N\times 5}$ is the predicted layouts for the $N$ objects in the image.

### 3.2 GENERATION-DETECTION CYCLE-CONSISTENT (GDCC) LEARNING FRAMEWORK

#### 3.2.1 TASK DUALITY AND CYCLE-CONSISTENCY

From §3.1, it becomes evident that L2I and OD can be viewed as inverse tasks of each other, where the input and output of L2I generation correspond to the output and input of OD, respectively. Though largely overlooked in previous research, such task duality can be effectively leveraged to improve the performance of both tasks through cycle consistency learning.

Specifically, if a layout is mapped to an image using an L2I generator $\mathcal{G}$, and then mapped back to a layout using an object detector $\mathcal{D}$, the process should recover the original layout. This forces consistency in what we term a **layout translation cycle**. In this cycle, $\mathcal{D}$ remains fixed while $\mathcal{G}$ is trained to minimize the discrepancy between the predicted and the original input layouts, ensuring more precise and realistic image generation that faithfully reflects the input layout. Similarly, mapping an image to a layout and then back again should ideally recover the original image. This ensures consistency in an **image translation cycle**. In this case, $\mathcal{G}$ is fixed, and $\mathcal{D}$ is trained to minimize the difference between the predicted and original images, thus enhancing its ability to accurately predict layouts from images. These two cycle-consistent learning processes improve both $\mathcal{G}$ and $\mathcal{D}$ in an end-to-end manner similar to GAN [17], with each receiving feedback from the other. In the following, we will present GDCC in both paired (§3.2.2) and unpaired (§3.2.3) data settings.

#### 3.2.2 GDCC IN PAIRED DATA SETTING

In the paired data setting, each image $\boldsymbol{x}_0 \in \mathbb{R}^{H\times W\times 3}$ is annotated with a structured layout $\boldsymbol{l} \in \mathbb{R}^{N\times 5}$ that includes bounding boxes and class labels for the objects in the image. The framework is shown in Fig. 2. Below, we detail the learning process of GDCC in this context.

**Layout Translation Cycle.** As discussed in §3.2.1, in this process, $\mathcal{D}$ remains fixed while $\mathcal{G}$ is trained to minimize the discrepancy between the predicted and the original input layouts to achieve more precise and realistic image generation that faithfully reflects the input layout.

Specifically, given an L2I generation model $\mathcal{G}$ and the layout input $\boldsymbol{l} \in \mathbb{R}^{N\times 5}$, a conditionally synthesized images $\boldsymbol{x}_1^{\mathrm{syn}} \in \mathbb{R}^{H\times W\times 3}$ can be obtained as follows:

$$\boldsymbol{x}_1^{\mathrm{syn}} = \mathcal{G}^T\big(t, \boldsymbol{\epsilon}, \boldsymbol{y}, \boldsymbol{l}\big). \tag{5}$$

Next, a pre-trained object detector $\mathcal{D}$ is employed to map $\boldsymbol{x}_1^{\mathrm{syn}}$ back into the layout space:

$$\hat{\boldsymbol{l}} = \mathcal{D}\big(\boldsymbol{x}_1^{\mathrm{syn}}\big), \tag{6}$$

Figure 2: **GDCC framework in paired data setting.** The L2I generator $\mathcal{G}$ maps from the layout space to the image space, while the object detector $\mathcal{D}$ performs the inverse mapping. Given a paired data with an input image $\boldsymbol{x}_0$ and its corresponding layout $\boldsymbol{l}$, $\mathcal{G}$ is trained with the layout translation cycle loss $\mathcal{L}_{\text{layoutTC}}$ and the diffusion model loss $\mathcal{L}_{\text{dm}}$, and $\mathcal{D}$ is trained with the image translation cycle loss $\mathcal{L}_{\text{imageTC}}$ and the prediction loss $\mathcal{L}_{\text{pred}}$. See §3.2.2 for details.

where a score threshold $s_{\text{thre}}$ is applied to filter the predicted bounding boxes, leading to a more stable training process. The **layout translation cycle loss** $\mathcal{L}_{\text{layoutTC}}$ is then computed by measuring the similarity between the input layout $\boldsymbol{l}$ and its dual layout $\hat{\boldsymbol{l}} \in \mathbb{R}^{N \times 5}$:

$$\mathcal{L}_{\text{layoutTC}} = \mathcal{L}_{\text{bbox}}(\boldsymbol{l}, \hat{\boldsymbol{l}}) = \mathcal{L}_{\text{reg}}\big(\{\boldsymbol{b}_n\}_{n=1}^N, \{\hat{\boldsymbol{b}}_n\}_{n=1}^N\big) + \mathcal{L}_{\text{cls}}\big(\{\boldsymbol{c}_n\}_{n=1}^N, \{\hat{\boldsymbol{c}}_n\}_{n=1}^N\big), \quad (7)$$

where the bounding box loss $\mathcal{L}_{\text{bbox}}$ consists of a smooth L1 loss $\mathcal{L}_{\text{reg}}$ for regression and a cross-entropy loss $\mathcal{L}_{\text{cls}}$ for classification.

**Perturbative Single-step Sampling.** The $T$-step samplings process to generate $\boldsymbol{x}_1^{\text{syn}}$ in Eq. (5) is time-consuming and requires gradient storage at each timestep to facilitate backpropagation, which reduces the efficiency of layout translation cycle. Inspired by [33], we implement a *perturbative single-step denoising strategy* to accelerate the L2I process. Instead of generating $\boldsymbol{x}_1^{\text{syn}}$ from Gaussian noise, we obtain a special noise $\boldsymbol{x}_t^{\text{pert}}$ by perturbing image $\boldsymbol{x}_0$ with a small noise $\boldsymbol{\epsilon}_0$ for $t \leq t_{\text{thre}}$ diffusion steps, where $t_{\text{thre}}$ is a hyper-parameter that constrains $\boldsymbol{\epsilon}_0$ to be relatively small. We then perform a single-step denoising process on $\boldsymbol{x}_t^{\text{pert}}$ to achieve L2I generation and obtain $\boldsymbol{x}_1^{\text{syn}}$:

$$\boldsymbol{x}_1^{\text{syn}} = \frac{\boldsymbol{x}_t^{\text{pert}} - \sqrt{1-\alpha_t}\,\boldsymbol{\epsilon}_\theta\big(t-1, \boldsymbol{x}_t^{\text{pert}}, \boldsymbol{y}, \boldsymbol{l}\big)}{\sqrt{\alpha_t}} = \mathcal{G}\big(t, \boldsymbol{x}_t^{\text{pert}}, \boldsymbol{y}, \boldsymbol{l}\big), \quad (8)$$

where $\mathcal{G}$ denotes the L2I generator that performs perturbative single-step denoising, which is guided by the diffusion model loss $\mathcal{L}_{\text{dm}}$ defined in Eq. (2). In summary, the total loss for training $\mathcal{G}$ in the layout transition cycle for the paired data setting is defined as follows:

$$\mathcal{L}_{\text{gen}} = \begin{cases} \mathcal{L}_{\text{dm}} + \lambda_1 \cdot \mathcal{L}_{\text{layoutTC}} & \text{if } t \leq t_{\text{thre}} \\ \mathcal{L}_{\text{dm}} & \text{otherwise} \end{cases}. \quad (9)$$

Here, $\lambda_1$ adjusts the weight of the layout translation cycle loss $\mathcal{L}_{\text{layoutTC}}$, and $t_{\text{thre}}$ denotes a threshold beyond which $\mathcal{L}_{\text{layoutTC}}$ is no longer applied, as the noise introduced in the perturbative single-step sampling process becomes too large to yield desired $\boldsymbol{x}_t^{\text{pert}}$ and $\boldsymbol{x}_1^{\text{syn}}$ for consistency learning.

**Image Translation Cycle.** As discussed in §3.2.1, in this process, $\mathcal{G}$ is fixed, and $\mathcal{D}$ is trained to minimize the difference between the predicted and original images, thereby improving its ability to accurately predict layouts from images.

Formally, the layout $\hat{\boldsymbol{l}}$ obtained from $\boldsymbol{x}_1^{\text{syn}}$ (*cf.*, Eq. (6)) can be remap to image space by $\mathcal{G}$, resulting in $\boldsymbol{x}_2^{\text{syn}} \in \mathbb{R}^{H \times W \times 3}$. The **image translation cycle loss** $\mathcal{L}_{\text{imageTC}}$ is then computed by evaluating the similarity between $\boldsymbol{x}_1^{\text{syn}}$ (*cf.* Eq. (8)) and $\boldsymbol{x}_2^{\text{syn}}$:

$$\begin{aligned} \mathcal{L}_{\text{imageTC}} &= \mathbb{E}_{t, \boldsymbol{x}_0, \boldsymbol{y}, \boldsymbol{l}, \boldsymbol{\epsilon} \sim \mathcal{N}(0,1)} \| \mathcal{G}\big(t, \boldsymbol{x}_t^{\text{pert}}, \boldsymbol{y}, \boldsymbol{l}\big) - \mathcal{G}\big(t, \boldsymbol{x}_t^{\text{pert}}, \boldsymbol{y}, \hat{\boldsymbol{l}}\big) \|_2^2 \\ &= \mathbb{E}_{t, \boldsymbol{x}_0, \boldsymbol{y}, \boldsymbol{l}, \boldsymbol{\epsilon} \sim \mathcal{N}(0,1)} \big\| \big[\boldsymbol{x}_t^{\text{pert}} - \sqrt{1-\bar{\alpha}_t}\,\boldsymbol{\epsilon}_\theta\big(t, \boldsymbol{x}_t^{\text{pert}}, \boldsymbol{y}, \boldsymbol{l}\big)\big]/\sqrt{\bar{\alpha}_t} \\ &\quad - [\boldsymbol{x}_t^{\text{pert}} - \sqrt{1-\bar{\alpha}_t}\,\boldsymbol{\epsilon}_\theta\big(t, \boldsymbol{x}_t^{\text{pert}}, \boldsymbol{y}, \hat{\boldsymbol{l}}\big)]/\sqrt{\bar{\alpha}_t} \big\|_2^2 \\ &= \mathbb{E}_{t, \boldsymbol{x}_0, \boldsymbol{y}, \boldsymbol{l}, \boldsymbol{\epsilon} \sim \mathcal{N}(0,1)} \big(\sqrt{(1-\bar{\alpha}_t)/\bar{\alpha}_t}\big) \|\boldsymbol{\epsilon}_\theta\big(t, \boldsymbol{x}_t^{\text{pert}}, \boldsymbol{y}, \boldsymbol{l}\big) - \boldsymbol{\epsilon}_\theta(t, \boldsymbol{x}_t^{\text{pert}}, \boldsymbol{y}, \hat{\boldsymbol{l}})\|_2^2. \end{aligned} \quad (10)$$

We obtain $\mathcal{L}_{\text{imageTC}} = \mathbb{E}_{t,\boldsymbol{x}_0,\boldsymbol{y},\boldsymbol{l},\boldsymbol{\epsilon}\sim\mathcal{N}(0,1)} \|\boldsymbol{\epsilon}_\theta(t,\boldsymbol{x}_t^{\text{pert}},\boldsymbol{y},\boldsymbol{l}) - \boldsymbol{\epsilon}_\theta(t,\boldsymbol{x}_t^{\text{pert}},\boldsymbol{y},\hat{\boldsymbol{l}})\|_2^2$ by omitting the scaling factor. As seen, with the above perturbative single-step denoising strategy, the image translation cycle only requires to compute the noise predicted by the U-Net denoiser $\boldsymbol{\epsilon}_\theta$ at timestep $t$ during two generation forward translations, which significantly improves the efficiency of GDCC.

To maintain the performance of $\mathcal{D}$ on real-world data, we make full use of the paired data by predicting the layout $\boldsymbol{l}_{\text{pred}} \in \mathbb{R}^{N\times5}$ from image $\boldsymbol{x}_0$, and minimizing the prediction loss between $\boldsymbol{l}_{\text{pred}}$ and the annotated layout $\boldsymbol{l}$, defined as $\mathcal{L}_{\text{pred}} = \mathcal{L}_{\text{bbox}}(\boldsymbol{l},\boldsymbol{l}_{\text{pred}})$, during the training of $\mathcal{D}$. In summary, the total loss for training $\mathcal{D}$ in the image translation cycle in paired data setting is as follows:

$$\mathcal{L}_{\text{det}} = \begin{cases} \mathcal{L}_{\text{pred}} + \lambda_2 \cdot \mathcal{L}_{\text{imageTC}} & \text{if } t \leq t_{\text{thre}} \\ \mathcal{L}_{\text{pred}} & \text{otherwise} \end{cases}. \tag{11}$$

Similar to Eq.(9), $\lambda_2$ is the weight of $\mathcal{L}_{\text{imageTC}}$. The image translation cycle is performed within $t_{\text{thre}}$ timesteps to fulfill the constraint of the perturbative single-step denoising strategy.

**Priority Timestep Re-Sampling.** In the training of DMs, a random timestep $t$ is selected from $1$ to $t_{\text{max}}$ at each training step, and the model is trained to predict the added noise at this particular timestep. However, in our experiment, since $t_{\text{thre}} \ll t_{\text{max}}$, the traditional uniform sampling strategy results in a low probability of selecting a $t \in [1, t_{\text{thre}}]$ to trigger the layout or image translation cycle loss in Eq. (9) or (11). This leads to slow convergence during training. To alleviate this issue, we propose a *priority timestep re-sampling strategy*, which applies a re-weighting factor $w > 1$ to prioritize the selection of $t \in [1, t_{\text{thre}}]$. The re-weighted timestep probability density function $p_{\text{reweight}}(t)$ is defined as follows:

$$p_{\text{reweight}}(t) = \begin{cases} w/t_{\text{thre}} & \text{if } t \leq t_{\text{thre}} \\ (1 - w \cdot t_{\text{thre}}/t_{\text{max}})/(t_{\text{max}} - t_{\text{thre}}) & \text{otherwise} \end{cases}. \tag{12}$$

This strategy increases the frequency of triggering layout and image translation cycle losses during training, thus accelerating convergence. The effectiveness of this re-sampling strategy is demonstrated by the results shown in Table 6b. When combined with the perturbative single-step denoising strategy introduced above, our GDCC becomes significantly more streamlined and efficient.

### 3.2.3 GDCC IN UNPAIRED DATA SETTING

In addition to leveraging large-scale annotated layout-image pairs to achieve mutual improvement of the L2I generator and object detector, GDCC also facilitates more efficient use of unpaired data, thereby further enhancing data efficiency. In this section, we explore GDCC learning with layouts as the sole training data. To obtain more unpaired layouts, we utilize VisorGPT [70], a recent generative pre-training model to automatically sample layouts based on its learned visual priors.

In the unparied data setting, the sampled layout $\boldsymbol{l}^{\text{syn}} \in \mathbb{R}^{N\times5}$ or the real-world layout $\boldsymbol{l}^{\text{real}} \in \mathbb{R}^{N\times5}$ functions identically to the layout input $\boldsymbol{l}$ in the paried data setting for the L2I generation described in Eq.(5). This allows for the calculation of the layout translation cycle loss $\mathcal{L}_{\text{layoutTC}}$ in Eq.(7) and image translation cycle loss $\mathcal{L}_{\text{imageTC}}$ in Eq.(10). However, due to the absence of the corresponding image $\boldsymbol{x}_0$, it becomes impossible to calculate $\mathcal{L}_{\text{ldm}}$ and $\mathcal{L}_{\text{pred}}$, and thus cannot apply the perturbative single-step sampling and priority timestep re-sampling strategies. To reduce the GPU memory in this situation, only a subset of the gradients is retained during the $T$-step samplings for L2I image generation. In summary, the training loss of $\mathcal{G}$ reduces to $\mathcal{L}_{\text{gen}} = \mathcal{L}_{\text{layoutTC}}$ and the training loss of $\mathcal{D}$ simplifies to $\mathcal{L}_{\text{det}} = \mathcal{L}_{\text{imageTC}}$ under the unpaired data setting for GDCC learning. Experiment results are presented in Table 5. Related details are shown in Appendix §D.

## 4 EXPERIMENTS

### 4.1 EXPERIMENTAL SETUP

Following [6], we train the models on the COCO-Stuff [3] training split and test on COCO 2017 [38], while for NuImages [4], we use its respective training and testing splits. For L2I generation models, *fidelity* is evaluated using Frechet Inception Distance (FID) and the YOLO score [36], while *trainability* is measured by the fine-tuning performance of object detection (OD) models using Average Precision (AP). For OD models, both generation trainability and detection fine-tuning performance are assessed using AP. Related details are shown in Appendix §B.

**Training.** We conduct experiments for GDCC using two L2I generators, namely GeoDiffusion [6] and ControlNet [75], and an object detector Faster R-CNN [50]. For GeoDiffusion, we conducted fine-tuning on COCO-Stuff [3] and NuImages [4], respectively. In this process, only the U-Net denoiser parameters are updated, while all other parameters remain fixed. The text prompt is replaced with a null text with

Table 1: **Quantitative results of generation fidelity on COCO 2017**. GDCC is fune-tuned on pre-trained L2I methods. †: re-implementation from GeoDiffusion [6]. ‡: with additional mask annotations. See §4.2.

| Method | Res. | Epoch | FID ↓ | mAP ↑ | AP$_{50}$ ↑ | AP$_{75}$ ↑ |
|---|---|---|---|---|---|---|
| LostGAN [59] [ICCV 19] | | 200 | 42.55 | 9.1 | 15.3 | 9.8 |
| LAMA [36] [ICCV 21] | | 200 | 31.12 | 13.4 | 19.7 | 14.9 |
| CAL2IM [21] [CVPR 21] | | 200 | 25.95 | 10.0 | 14.9 | 11.1 |
| Taming [27] [ArXiv 21] | | 128 | 33.68 | - | - | - |
| TwFA [74] [CVPR 22] | $256^2$ | 300 | 22.15 | - | 28.2 | 20.1 |
| Frido [15] [AAAI 23] | | 200 | 37.14 | 17.2 | - | - |
| L.Diffusion† [77] [CVPR 23] | | 180 | 22.65 | 14.9 | 27.5 | 14.9 |
| DetDiffusion‡ [67] [CVPR 24] | | 60 | 19.28 | 29.8 | 38.6 | 34.1 |
| GeoDiffusion [6] [ICLR 24] | | 60 | 20.16 | 29.1 | 38.9 | 33.6 |
| GeoDiffsion − **GDCC** | | 2 | **18.09** $_{\pm0.11}$ | **31.2** $_{\pm0.1}$ | **41.1** $_{\pm0.1}$ | **36.2** $_{\pm0.2}$ |
| ReCo† [73] [CVPR 23] | | 100 | 29.69 | 18.8 | 33.5 | 19.7 |
| L.Diffuse† [9] [ArXiv 23] | | 60 | 22.20 | 11.4 | 23.1 | 10.1 |
| GLIGEN [35] [CVPR 23] | | 86 | 21.04 | 22.4 | 36.5 | 24.1 |
| ControlNet [75] [ICCV 23] | $512^2$ | 60 | 28.14 | 25.2 | 46.7 | 22.7 |
| ControlNet − **GDCC** | | 2 | 26.68 $_{\pm0.09}$ | 26.9 $_{\pm0.2}$ | 47.8 $_{\pm0.1}$ | 24.0 $_{\pm0.2}$ |
| GeoDiffusion [6] [ICLR 24] | | 60 | 18.89 | 30.6 | 41.7 | 35.6 |
| GeoDiffsion − **GDCC** | | 2 | **17.36** $_{\pm0.09}$ | **32.5** $_{\pm0.1}$ | **43.5** $_{\pm0.1}$ | **38.0** $_{\pm0.2}$ |

a probability of 0.1 to allow unconditional generation following [6]. We adopt AdamW [26] with a momentum of 0.9 and a weight decay of 0.01. The learning rate is set to $3 \times 10^{-5}$, and adjusted using a cosine schedule [42] with a 3,000-iteration warm-up. The batch size is 56. GeoDiffusion is fine-tuned for 2 epochs on COCO-Stuff and 3 epochs on NuImages, which is remarkably efficient.

For ControlNet, as the official implementation does not support bounding boxes as conditional inputs, we first convert bounding boxes into masks for conditional input and train on COCO-Stuff accordingly. Then, we finetune the pretrained ControlNet using GDCC for 2 epochs by updating only the ControlNet-specific parameters and keep all others frozen.

Faster R-CNN [50], pre-trained separately on the COCO 2017 and the NuImages training sets, is employed for the respective datasets. A score threshold $s_{\text{thre}} = 0.5$ is used to filter the predicted bounding boxes. Each predicted bounding box is assigned to a ground truth box with an Intersection over Union (IoU) of at least 0.5, or classified as background.

Table 2: **Quantitative results of detection fine-tuning and generation trainability on COCO 2017**. A Faster R-CNN pre-trained on COCO 2017 is employed as the baseline. Detection fine-tuning refers to fine-tuning the detector during the training of GDCC, while generative trainability denotes the re-training of the detector on generated and real samples. The input resolution is set to 800×456 following [6]. See §4.2.

| Method | mAP ↑ | AP$_{50}$ ↑ | AP$_{75}$ ↑ | AP$^m$ ↑ | AP$^l$ ↑ |
|---|---|---|---|---|---|
| – Detection Fine-tuning – | | | | | |
| Faster R-CNN [50] [NIPS 15] | 37.3 | 58.2 | 40.8 | 40.7 | 48.2 |
| Faster R-CNN – **GDCC** | **38.2** $_{\pm0.1}$ | **58.5** $_{\pm0.1}$ | **41.9** $_{\pm0.1}$ | **41.5** | **49.0** |
| – Generation Trainability – | | | | | |
| L.Diffusion [77] [CVPR 23] | 36.5 | 57.0 | 39.5 | 39.7 | 47.5 |
| L.Diffuse [9] [ArXiv 23] | 36.6 | 57.4 | 39.5 | 40.0 | 47.4 |
| GLIGEN [35] [CVPR 23] | 36.8 | 57.6 | 39.9 | 40.3 | 47.9 |
| ControlNet [75] [ICCV 23] | 36.9 | 57.8 | 39.6 | 40.4 | 49.0 |
| GeoDiffusion [6] [ICLR 24] | 38.4 | 58.5 | 42.4 | 42.1 | 50.3 |
| GeoDiffsion – **GDCC** | **38.9** $_{\pm0.1}$ | **58.9** $_{\pm0.1}$ | **43.0** $_{\pm0.2}$ | **42.6** | **50.6** |

We adopt an alternating fine-tuning strategy for training L2I and OD models. In each epoch, the L2I model is trained for 1,000 iterations, followed by 1,000 iterations for the OD model. In the paired data setting, we set $\lambda_1 = \lambda_2 = 0.1$, $t_{\text{thre}} = 50$ and $t_{\text{max}} = 1000$ for Eqs.(9),(11) and (12), respectively.

**Testing.** Our GDCC framework preserves the original architectures of all the L2I and OD models, ensuring that the inference speed of each model remains unchanged. During image sampling, PLMS scheduler [41] is used to sample images from the NuImages dataset layouts for 100 steps with classifier-free guidance (CFG) scale of 5.0, and from the COCO-Stuff [3] dataset layouts for 50 steps with a CFG scale of 4.5. Following GeoDiffusion [6], for NuImages dataset [4], fidelity is assessed using a Mask R-CNN [20] object detector pre-trained on the NuImages training set to achieve a comparable YOLO score in LAMA [36]. For evaluation on COCO-Stuff, we use YOLOv4 [1] per-trained on COCO 2017 training set.

Table 3: **Quantitative results of generation fidelity on NuImages**. GDCC is fune-tuned on pre-trained L2I methods. "pred." denotes pedestrian. See §4.2 for details.

| Method | Res. | Epoch | FID ↓ | Average Precision↑ | | | | | | | |
| --- | --- | --- | --- | --- | --- | --- | --- | --- | --- | --- | --- |
| | | | | mAP | AP$_{50}$ | AP$_{75}$ | AP$^m$ | AP$^l$ | trailer | ped. | car |
| Oracle | - | - | - | 48.2 | 75.0 | 52.0 | 46.7 | 60.5 | 17.8 | 48.5 | 64.9 |
| LostGAN [59] [ICCV 19] | | 256 | 59.95 | 4.4 | 9.8 | 3.3 | 2.1 | 12.3 | 0.3 | 2.7 | 12.2 |
| LAMA [36] [ICCV 21] | | 256 | 63.85 | 3.2 | 8.3 | 1.9 | 2.0 | 9.4 | 1.4 | 1.3 | 8.8 |
| Taming [27] [ArXiv 21] | 256² | 256 | 32.84 | 7.4 | 19.0 | 4.8 | 2.8 | 18.8 | 6.0 | 3.0 | 17.3 |
| GeoDiffusion [6] [ICLR 24] | | 64 | 14.58 | 15.6 | 31.7 | 13.4 | 6.3 | 38.3 | 13.3 | 6.5 | 26.3 |
| GeoDiffsion − GDCC | | 3 | **12.76** ±0.13 | **17.4** ±0.1 | **33.5** ±0.1 | **15.5** ±0.2 | **8.2** | **40.3** | **14.8** | **8.0** | **28.5** |
| ReCo [73] [CVPR 23] | | 64 | 27.10 | 17.1 | 41.1 | 11.8 | 10.9 | 36.2 | 8.0 | 7.6 | 31.8 |
| GLIGEN [35] [CVPR 23] | | 64 | 16.68 | 21.3 | 42.1 | 19.1 | 15.9 | 40.8 | 8.5 | 14.7 | 38.7 |
| ControlNet [75] [ICCV 23] | 512² | 64 | 23.26 | 22.6 | 43.9 | 20.7 | 17.3 | 41.9 | 10.5 | 16.7 | 40.7 |
| GeoDiffusion [6] [ICLR 24] | | 64 | 9.58 | 31.8 | 62.9 | 28.7 | 27.0 | 53.8 | 21.2 | 18.2 | 46.0 |
| GeoDiffsion − GDCC | | 3 | **8.03** ±0.11 | **33.4** ±0.2 | **64.6** ±0.2 | **30.6** ±0.1 | **28.6** | **55.7** | **29.4** | **20.2** | **47.5** |

The pre-trained detector first performs inference on the generated images, and the resulted predictions are then compared with the corresponding ground truth annotations. Following [6], Frechet Inception Distance (FID) [23] is computed by generating five images for COCO-Stuff and one image for NuImage to calculate the distance between generated images and authentic images. All images are resized into $256 \times 256$ before evaluation. To assess the trainability, we augment the original training data with generated images and their corresponding layouts, creating a unified dataset. We then train Faster R-CNN [50] on this unified dataset using the standard $1\times$ schedule. The model employs ResNet-50 [19] pre-trained on ImageNet-1K [12] as its backbone and FPN [39] as the neck. The trained detection models are evaluated on validation set.

**Reproducibility.** GDCC is implemented in PyTorch. We use four NVIDIA V100 GPUs for training and a single NVIDIA A100 GPU for testing. Our reported results are averaged over three runs. To ensure reproducibility, our code will be released.

Table 4: **Quantitative results of detection fine-tuning and generation trainability on NuImages**. A pre-trained Faster R-CNN detector is employed as baseline. See §4.2.

| Method | | mAP ↑ |
| --- | --- | --- |
| – Detection Fine-tuning – | | |
| Faster R-CNN [50] [NIPS 15] | | 36.9 |
| Faster R-CNN − GDCC | | **37.7** ±0.1 |
| – Generation Trainability – | | |
| LostGAN | [59] [ICCV 19] | 35.6 |
| LAMA | [36] [ICCV 21] | 35.6 |
| Taming | [27] [ArXiv 21] | 35.8 |
| ReCo | [73] [CVPR 23] | 36.1 |
| GLIGEN | [35] [CVPR 23] | 36.3 |
| ControlNet | [75] [ICCV 23] | 36.4 |
| GeoDiffusion [6] [ICLR 24] | | 38.3 |
| GeoDiffusion − GDCC | | **38.7** ±0.1 |

## 4.2 QUANTITATIVE RESULTS

**Generation Fidelity on COCO 2017** [38]. The quality of generation is predicated on two key criteria: fidelity and trainability. For generation fidelity, as shown in Table 1, our GDCC learning framework significantly improves existing L2I generation methods in terms of both image fidelity, as measured by FID, and control fidelity, as evaluated by YOLO score, by a large degree.

At a $256 \times 256$ input resolution, for the GeoDiffusion [6] method, our GDCC framework achieves improvements of **2.1%/2.2%/2.6%** in mAP, mAP$_{50}$, and mAP$_{75}$, reaching **31.2%/41.1%/36.2%**, even surpassing the performance of original GeoDiffusion at a $512 \times 512$ resolution. Additionally, it achieves a **2.07%** improvement in FID. It is worth noting that, despite DetDiffusion [67] employing additional and detailed mask annotations for supervision while GDCC only uses bounding box label, our method still outperforms it. For a $512 \times 512$ input, GDCC also achieves **1.9%/1.8%/2.4%** mAP and **1.53%** FID enhancement compared with initial model, demonstrating the **state-of-the-art** performance in L2I generation realm. Based on the classic control generation method ControlNet [75], the GDCC learning framework achieves notable enhancements as well.

The enhanced FID and YOLO score achieved with GDCC demonstrate its effectiveness. GDCC not only enables precise layout control in generation but also enhances quality of the generated images, improving their resemblance to real-world data. Additionally, the improvements across different controllable generation methods demonstrate that GDCC is not dependent on any specific approach, highlighting its robustness and extensibility. Furthermore, GDCC is fine-tuned for only 2 epochs based on the pre-trained diffusion model, while the original implementation requires 60 epochs to reach convergence.

**Detection Performance and Generation Trainability on COCO 2017** [38]. A Faster R-CNN detector [50] trained on the COCO 2017 training set is employed for detection fine-tuning. To begin with, we set the performance of the detector on COCO 2017 validation set as our baseline.

As can be seen in Table 2, fine-tuning the detector at GDCC training process in an end-to-end manner leads to performance improvements across all metrics on the validation set. **For the first time**, we demonstrate that the L2I generation model can be advantageous to the object detector during training in an end-to-end manner, while previous works [6; 67] only use generated images to re-train the detector after training L2I generation model. In order to make a comparison of generation trainability, we also train the detector with generated and real images with ImageNet [12] pre-trained weights. As shown, GeoDiffusion fine-tuned with GDCC achieves **1.6 %/0.7%/1.2%** AP improvement over the baseline, outperforming the original GeoDiffusion performance.

**Generation Fidelity on NuImages** [4]. To illustrate the generalizability of GDCC with respect to dataset, more experiments are conducted on NuImages. As presented in Table 3, GDCC outperforms all baselines in FID and YOLO score after three epochs of fine-tuning.

**Detection Performance and Generation Trainability on NuImages** [4]. As can be seen in Table 4, GDCC achieves improvement on NuImages validation set after fine-tuning Faster-RCNN which is pre-trained on training set. In a data augmentation manner, GDCC demonstrates an accuracy improvement of **1.8%** compared to the baseline.

**Performance in unpaired data setting on COCO 2017** [38]. Under the condition where only layouts are available, as demonstrated in Table 5, our GDCC still exhibits a performance enhancement. With the synthesized layouts sampled from generative pre-training models [70], GDCC outperforms the baseline, demonstrating its data efficiency. By incorporating real-world layouts from COCO annotations, performance can be further enhanced. Related details are shown in Appendix §D.

Table 5: **Quantitative results in unpaired setting on COCO 2017**. Here, "syn", "real", and "union" denote synthesized, real-world, and combined layouts, respectively. See §4.2.

| Methods | Detection Score ↑ | Generation Fidelity | |
|---|---|---|---|
| | | FID ↓ | YOLO score ↑ |
| Baseline | 37.3 | 20.16 | 29.1 |
| – unpaired layout data – | | | |
| syn | 37.5 | 19.74 | 29.5 |
| real | 37.5 | 19.46 | 29.7 |
| union | 37.6 | 19.28 | 29.9 |
| – paired layout-image data – | | | |
| paired | 38.2 | 18.09 | 31.2 |

### 4.3 QUALITATIVE RESULTS

Fig. 3 shows representative generation visual results on COCO 2017, with the same random seed used during sampling to ensure fair comparison. L2I model [6] demonstrates stronger layout controllability (1st and 2nd columns) and superior image fidelity (2nd column) after fine-tuning with GDCC. Fig. 4 presents the detection results. As seen, after fine-tuning with GDCC , Faster R-CNN [50] demonstrates advanced detection performance as well.

### 4.4 DIAGNOSTIC EXPERIMENTS

To gain more insights into GDCC, we conduct a set of ablative studies on COCO 2017 [38] using GeoDiffusion [6] with a resolution of 256×256 as the baseline.

**Essential Components.** As shown in Table 6a, the diffusion training loss $\mathcal{L}_{dm}$ (*cf.* Eq. (2)) and the prediction loss $\mathcal{L}_{pred}$ lead to a slight improvement in generation fidelity and detection score, respectively, due to more iterations on training samples. When fine-tuning the generator with $\mathcal{L}_{gen}$ (*cf.* Eq. (9)) which contains both $\mathcal{L}_{dm}$ and layout translation cycle loss $\mathcal{L}_{layoutTC}$ (*cf.* Eq. (7)), there is a significant improvement in generation fidelity. Similarity, $\mathcal{L}_{det}$ (*cf.* Eq. (11)) with image translation cycle loss $\mathcal{L}_{imageTC}$ (*cf.* Eq. (10)) further improve the detector performance. Full GDCC, fine-tuning both the generator and detector in an end-to-end manner, achieves superior performance on both generation and detection metrics compared with each individual component. This clearly demonstrates the duality of two tasks, and GDCC facilitates mutual enhancement between them.

**Cycle Consistency.** As shown in Table 6b, setting $t_{thre} = 0$ indicates that no cycle-consistent loss is applied, and only $\mathcal{L}_{ldm}$ and $\mathcal{L}_{pred}$ are active. For $t_{thre} = 50$ without priority timestep resampling, generation fidelity improves thanks to the layout translation cycle. A notable performance boost is observed with $w = 6$, showing the effectiveness of priority timestep re-sampling strategy. However, further increasing $t_{thre}$ or $w$ results in a decline in performance, indicating that excessive noise disturbance or imbalanced sampling strategy can cause instability during training.

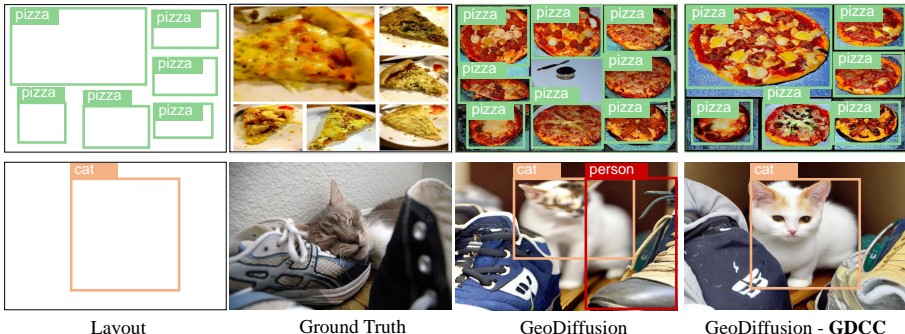

**Figure 3: Generation visual results of GeoDiffusion – GDCC on COCO 2017**. For fair comparisons, same seed is employed for sampling. See §4.3 for details.

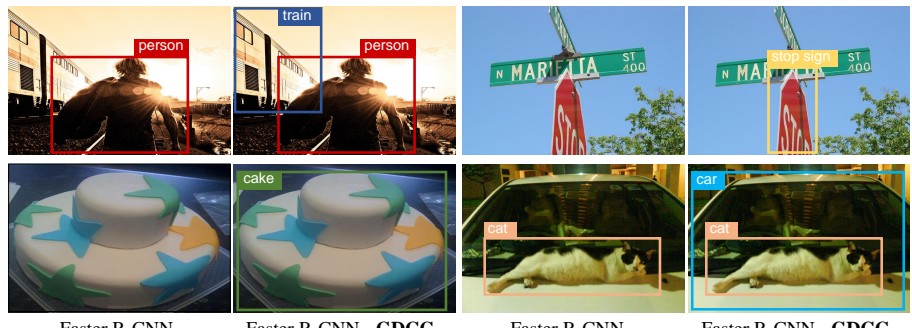

**Figure 4: Detection visual results of Faster R-CNN – GDCC on COCO 2017**. See §4.3 for details.

**Different Detectors.** In our main experiments, we deploy Faster R-CNN [50] as the detector. To investigate the robustness of GDCC across different detectors, experiments on Mask R-CNN [20] and Cascade R-CNN [5] are conducted. As illustrated in Table 6c, GDCC improves both the detection and generation score with different detectors. Furthermore, the stronger the performance of the detector, the more substantial the improvement in generation fidelity, reflecting the task duality between detection and generation again.

Table 6: **A set of ablative experiments on COCO 2017**. GeoDiffusion [6] with 256×256 resolution pre-trained on COCO-Stuff [3] is employed as L2I baseline. See §4.4 for details.

| Components | Detection Score ↑ | Generation Fidelity FID ↓ | YOLO score ↑ |
|---|---|---|---|
| Baseline | 37.3 | 20.16 | 29.1 |
| + $\mathcal{L}_{ldm}$ | 37.3 | 20.01 | 29.3 |
| + $\mathcal{L}_{gen}$ | 37.3 | 18.94 | 30.4 |
| + $\mathcal{L}_{pred}$ | 37.4 | 20.16 | 29.1 |
| + $\mathcal{L}_{det}$ | 37.7 | 20.16 | 29.1 |
| GDCC | **38.2** | **18.09** | **31.2** |

(a) essential components

| $t_{thre}$ | $w$ | FID ↓ | YOLO score↑ |
|---|---|---|---|
| 0 | 0 | 19.96 | 29.5 |
| 50 | 0 | 19.57 | 30.1 |
| 50 | 3 | 18.98 | 30.7 |
| 50 | 6 | **18.09** | **31.2** |
| 100 | 3 | 18.25 | 30.6 |
| 100 | 6 | 19.28 | 30.5 |
| 200 | 2 | 19.46 | 30.3 |

(b) reward strategy

| Detectors | Detection Score ↑ original | Detection Score ↑ fine-tuning | Generation Fidelity FID ↓ | YOLO score ↑ |
|---|---|---|---|---|
| Faster R-CNN [50] [NIPS 15] | 37.3 | 38.2 | 18.09 | 31.2 |
| Mask R-CNN [20] [ICCV 17] | 38.2 | 40.0 | 18.07 | 31.5 |
| Cascade R-CNN [5] [CVPR 18] | 40.3 | 41.2 | **18.04** | **31.7** |

(c) different detectors

## 5 CONCLUSION

In this paper, we propose GDCC, an end-to-end framework that jointly optimizes L2I generation and OD tasks. By exploring the inherent duality between these two tasks, GDCC facilitates mutual enhancement of L2I and OD models through the layout and image translation cycle losses. Additionally, GDCC allows for more efficient use of unpaired layout data, thereby further enhancing data efficiency. Notably, our GDCC is computationally efficient thanks to the perturbative single-step sampling and priority timestep re-sampling strategies during training, while maintaining the same inference cost as the original L2I and OD models. Extensive experiments confirm that GDCC significantly improves the controllability of diffusion-based L2I models and the accuracy of OD models.

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

- §A discusses our limitations, directions of our future work, and societal impact.
- §B introduces the datasets and evaluation metrics used in our experiments.
- §C provides the pseudo code of GDCC.
- §D presents more detailed discussions of GDCC under the unpaired setting.
- §E offers more detailed discussions regarding the fine-tuning performance and training cost.
- §F depicts more qualitative results of generation.
- §G provides more qualitative results of detection.

## A  LIMITATION, FUTURE WORK, AND SOCIAL IMPACT

**Limitation and Future Work.** In this work, we explore the inherent duality between layout-to-image (L2I) generation and object detection (OD). However, due to restrictions in computational resources, this duality is not extended to a broader range of controllable T2I generation and discriminative models, such as segmentation mask controllable models paired with segmentation models, and depth map controllable models paired with depth models, *etc.*. In future work, we aspire to expand the end-to-end joint learning framework for broader controllable T2I generation and discriminative models. In addition, our experiments in Table 2 and Table 8 also suggest that our highly realistic generated images aligned with synthesized layouts can benefit the training of object detectors. Therefore, another essential future direction deserving of further investigation is the construction of a large-scale synthetic dataset comprising synthesized layouts and their corresponding images generated by advanced L2I generation models. Overall, we believe the results presented in this paper warrant further exploration.

**Social Impact.** This work investigates the inherent duality between the L2I generation and OD and introduces GDCC learning framework that jointly optimizes both two tasks in an end-to-end manner. On positive side, the approach advances both L2I generation and OD model accuracy, leading to more precise scene synthesis and object localization. Improved L2I generation model can generate realistic images consistent with layouts, benefiting fields such as content creation and synthesized dataset construction. Meanwhile, the enhanced OD model offers advantages in areas like autonomous driving and surveillance systems. For potential negative social impact, the ability to generate highly realistic images could be misused to produce misleading or fake content, raising significant ethical concerns around surveillance, privacy, and the potential for digital manipulation.

## B  DATASETS AND EVALUATION METRICS

**Datasets.** Our experiments are conducted on two widely used datasets. *COCO-Stuff* [3] consists of bounding box annotations covering 80 object classes and 91 stuff classes. Following [28; 36; 6], objects occupying less than 2% of the total image area are ignored, and only images with 3 to 8 objects are used, resulting in a dataset of 74,777 training images and 3,097 validation images. We train on the COCO-Stuff training split and test on COCO 2017 following [6]. *NuImages* [4] offers bounding box annotations across 10 categories and 6 camera views. We exclude images with more than 22 objects following [6], yielding 60,209 images for training and 14,772 images for validation.

**Evaluation Metric.** L2I generation models are evaluated using two main criteria: *fidelity* and *trainability*. Fidelity assesses the consistency between the generated object representations and the authentic distribution of images. Specifically, fidelity quality is measured using the Frechet Inception Distance (FID) [23] from the perceptual perspective, while YOLO score proposed by [36] is used to evaluate the alignment between the generated images and conditional layouts.

## C PSEUDO CODE OF GDCC AND CODE RELEASE

The pseudo-code of GDCC is given in Algorithm 1. To guarantee reproducibility, our full implementation shall be publicly released upon acceptance.

**Algorithm 1** Pseudo-code of GDCC in a PyTorch-like style.

```
"""
vae: mapping to latent space
scheduler: adding noise to an image or for updating a sample
unet: predicting the noise
detector: object detector
x: input image (B x 3 x H x W)
l: input layout (B x 5)
t: input text description (B x L)
encoder_hidden_states: output of text_encoder(t)
noise: random sampled Gaussian noise
max_ts: max timestep for reward
resample_ts: re-weighting factor for timestep reward
reward_scale: balance reward loss and original loss
"""

# fine-tune generation model
if train_unet:
    unet.train()
    unet.requires_grad_(True)
    detector.eval()
    detector.requires_grad_(False)

    # Convert images to latent space
    latents = vae.encode(x)

    # Sample timesteps for each image
    timesteps = sample_timesteps(num_train_timesteps, max_ts, resample_ts)
    # Determine which samples need to calculate reward loss
    timestep_mask = (timesteps <= max_ts)

    # Add noise to the latents according to the noise at each timestep
    noisy_latents = scheduler.add_noise(latents, noise, timesteps)

    # Predict the noise residual and compute loss
    noise_pred = unet(noisy_latents, timesteps, encoder_hidden_states, l).sample

    # Predict the single-step denoised latents
    sample_latents = scheduler.step(noise_pred, timesteps, noisy_latents).
        pred_original_sample

    # Reconstruct images according to the predicted noise (Eq. 8)
    reconstructed_images = vae.decode(sample_latents).sample

    # Detect the reconstructed images and get dual layouts with logits
    # A threshold is adopted to filter bboxes (Eq. 6)
    dual_l, logits = detector(reconstructed_images)

    # Compute the layout translation loss (Eq. 9)
    box_loss, cls_loss = calculate_box_loss(dual_l, logits, l)
    l_cycle_loss = box_loss + cls_loss

    # Original Latent Diffusion Loss (Eq. 2)
    ldm_loss = ((noise_pred - noise) ** 2).mean()

    # total training loss for the generatino model (Eq. 10)
    l_cycle_loss = l_cycle_loss * timestep_mask.sum() / timestep_mask.sum()
    gen_loss = ldm_loss + l_cycle_loss * reward_scale

    # Optimize the generation model
    gen_loss.backward()
    optimizer.step()
    optimizer.zero_grad()
```

```python
    # fine-tune detection model
    else:
        unet.eval()
        unet.requires_grad_(False)
        detector.train()
        detector.requires_grad_(Ture)

        # Convert images to latent space
        latents = vae.encode(x)

        # Sample timesteps for each image
        timesteps = sample_timesteps(num_train_timesteps, max_ts, resample_ts)
        # Determine which samples need to calculate reward loss
        timestep_mask = (timesteps <= max_ts)

        # Add noise to the latents according to the noise at each timestep
        noisy_latents = scheduler.add_noise(latents, noise, timesteps)

        # Predict the noise residual and compute loss
        noise_pred = unet(noisy_latents, timesteps, encoder_hidden_states, l).sample

        # Predict the single-step denoised latents
        sample_latents = scheduler.step(noise_pred, timesteps, noisy_latents).
            pred_original_sample

        # Reconstruct images according to the predicted noise (Eq. 8)
        reconstructed_images = vae.decode(sample_latents).sample

        # Detect the reconstructed images and get dual layouts with logits
        # A threshold is adopted to filter bboxes (Eq. 6)
        dual_l, logits = detector(reconstructed_images)

        # Compute the image translation loss (Eq. 11)
        noise_pred_2 = unet(noisy_latents, timesteps, encoder_hidden_states, dual_l).sample
        i_cycle_loss = ((noise_pred - noise_pred_2) ** 2).mean()

        # Compute the prediction loss (Eq. 12)
        pred_l, logits = detector(x)
        pred_box_loss, pred_cls_loss = calculate_box_loss(pred_l, logits, l)
        pred_loss = pred_box_loss + pred_cls_loss

        det_loss = pred_loss + i_cycle_loss * reward_scale

        # Optimize the generation model
        det_loss.backward()
        optimizer.step()
        optimizer.zero_grad()

def sample_timesteps(num_train_timesteps, max_ts, resample_ts):
    # Initialize timestep
    timesteps = torch.arange(0, num_train_timesteps)
    probs = torch.ones(total_timesteps, device='cuda')

    # Reward re-weighting (Eq. 13)
    reward_indices = (timesteps <= max_ts)
    probs[reward_indices] *= resample_ts

    # Normalize probability distribution
    probs = probs / probs.sum()

    # Sample according to the weights
    sampled_timesteps = torch.multinomial(probs, bsz, replacement=True)

    return sampled_timesteps
```

# D    DISCUSSIONS REGARDING THE UNPAIRED SETTING OF GDCC

Our GDCC demonstrates efficiency in both paired layout-image and unpaired layout settings. In this section, we focus on the unpaired setting and provide detailed discussions.

**Experimental Setup.** As mentioned in §3.2.3, we adopt VisorGPT [70], a recent generative pre-training model to automatically sample layouts based on its learned visual priors. More specifically, VisorGPT requires users to input the object names and the number of instances for each image to generate layouts. we first sample synthesized layouts by inputting the class names and the number of instances from each image in the COCO 2017 [38] training set into VisorGPT. This process allows us to obtain corresponding ground truth layouts and synthesized layouts with the same number. To investigate the impact of varying the number of synthesized layouts on performance, we also experiment by randomly increasing or decreasing the number of instances in the COCO annotations, as well as altering the random seed to generate new synthesized layouts. In the end, we obtained three different ratios of synthesized layouts to ground truth layouts: $1/2$, 1, and 2.

In the following sections, we present two main experiments. The first involves fine-tuning both the generation model and detection model using the end-to-end GDCC learning framework on the synthesized and ground truth layouts, similar to Table 5. The second experiment focuses on re-training the detection model using the synthesized data, akin to the generation trainability experiment in Table 2, to evaluate whether these synthesized layouts can further enhance the performance of the detection model.

**Additional Results of Fine-tuning in Unpaired Setting on COCO** [3]. As shown in Table 7, relying solely on synthesized layouts yields a modest performance improvement, albeit not as substantial as when using real-world layouts alone. The utilization of combined layouts results in performance improvements, with the optimal outcome observed when the ratio of synthesized to real-world layouts is balanced at 1:1. This suggests that increasing the proportion of synthesized layouts beyond this ratio does not lead to further performance improvements. Additionally, performance in the unpaired setting consistently lags behind that of the paired setting.

Table 7: **More quantitative results of fine-tuning in unpaired setting on 2017**. "syn", "real", and "union" denote synthesized layout, real-world layouts, and union layouts that encompass both, respectively. "Synthesized Ratio" represents the ratio of synthesized layouts to ground truth layouts. See §D for details.

| Setting | Synthesized Ratio | Detection Score ↑ | Generation Fidelity | |
|---|---|---|---|---|
| | | | FID ↓ | YOLO score ↑ |
| Baseline | - | 37.3 | 20.16 | 29.1 |
| – unpaired layout data – | | | | |
| syn | 0.5 | 37.3 | 20.11 | 29.2 |
| syn | 1 | 37.5 | 19.74 | 29.5 |
| syn | 2 | 37.4 | 20.02 | 29.4 |
| real | - | 37.5 | 19.46 | 29.7 |
| union | 0.5 | 37.5 | 19.37 | 29.6 |
| union | 1 | **37.6** | **19.28** | **29.9** |
| union | 2 | 37.5 | 19.31 | 29.7 |
| – paired layout-image data – | | | | |
| paired | - | 38.2 | 18.09 | 31.2 |

**Additional Results of Generation Trainability in Unpaired Setting on COCO** [38]. As illustrated in Table 8, we observe that when the quality of generated images is sufficiently high, increasing the number of synthesized layouts and re-training the object detector on a dataset that combines both real-world and synthesized data can further improve detection performance. This prompts us to construct a larger synthetic dataset, incorporating more data sampled from powerful L2I generation models to further enhance the performance of the detector. We leave it for future work.

Table 8: **More quantitative results of generation trainability on COCO 2017**. "Syn. Ratio" represents the ratio of synthesized layouts to ground truth layouts. A Faster R-CNN detector [50] pre-trained on COCO is employed as the baseline. The input resolution of generation model is set as 800×456 following [6]. See §D for details.

| Method | Syn. Ratio | mAP ↑ | $AP_{50}$ ↑ | $AP_{75}$ ↑ | $AP^m$ ↑ | $AP^l$ ↑ |
|---|---|---|---|---|---|---|
| Baseline | - | 37.3 | 58.2 | 40.8 | 40.7 | 48.2 |
| L.Diffusion [77] [CVPR 23] | 0 | 36.5 | 57.0 | 39.5 | 39.7 | 47.5 |
| L.Diffuse [9] [ArXiv 23] | 0 | 36.6 | 57.4 | 39.5 | 40.0 | 47.4 |
| GLIGEN [35] [CVPR 23] | 0 | 36.8 | 57.6 | 39.9 | 40.3 | 47.9 |
| ControlNet [75] [ICCV 23] | 0 | 36.9 | 57.8 | 39.6 | 40.4 | 49.0 |
| GeoDiffusion [6] [ICLR 24] | 0 | 38.4 | 58.5 | 42.4 | 42.1 | 50.3 |
| GeoDiffusion [6] [ICLR 24] | 1 | 38.7 | 58.7 | 42.7 | 42.3 | 50.7 |
| GeoDiffsion – GDCC | 0 | 38.9 | 58.9 | 43.0 | 42.6 | 50.6 |
| GeoDiffsion – GDCC | 1 | 39.4 | 59.3 | 43.6 | 43.0 | 51.1 |
| GeoDiffsion – GDCC | 2 | **39.7** | **59.5** | **44.0** | **43.2** | **51.3** |

# E    DISCUSSIONS REGARDING THE FINE-TUNING PERFORMANCE AND TRAINING COST

As illustrated in Table 9, we compare the fine-tuning performance and training cost with and without GDCC. Although GDCC increases training time by 0.7 hours and GPU memory usage by 11 GB with 2 epochs of fine-tuning, it achieves remarkable performance improvements of **2.07**% in FID, **2.1**% in YOLO score, and an additional **0.9**% in detection score. However, the performance after fine-tuning remains nearly unchanged without GDCC. This clearly demonstrates the effectiveness of our method, as significant performance improvements are not achieved through additional training epochs.

Table 9: **Fine-tuning performance and training cost on COCO 2017**. All models are tested on two Nvidia A100 GPUs with batch size 32 for each GPU. See §E for details.

| Methods | Epoch | Training Cost ↓ | | Generation Fidelity | |
|---|---|---|---|---|---|
| | | training hours | GPU memory | FID ↓ | YOLO score ↑ |
| – original – | | | | | |
| GeoDiffusion | 60 | 54.0 | 27G | 20.16 | 29.1 |
| – fine-tune – | | | | | |
| GeoDiffusion | 2 | 1.9 | 27G | 20.13 | 29.3 |
| GeoDiffusion - GDCC | 2 | 2.6 | 38G | **18.09** | **31.2** |

## F  MORE QUALITATIVE RESULTS OF GENERATION WITH GDCC

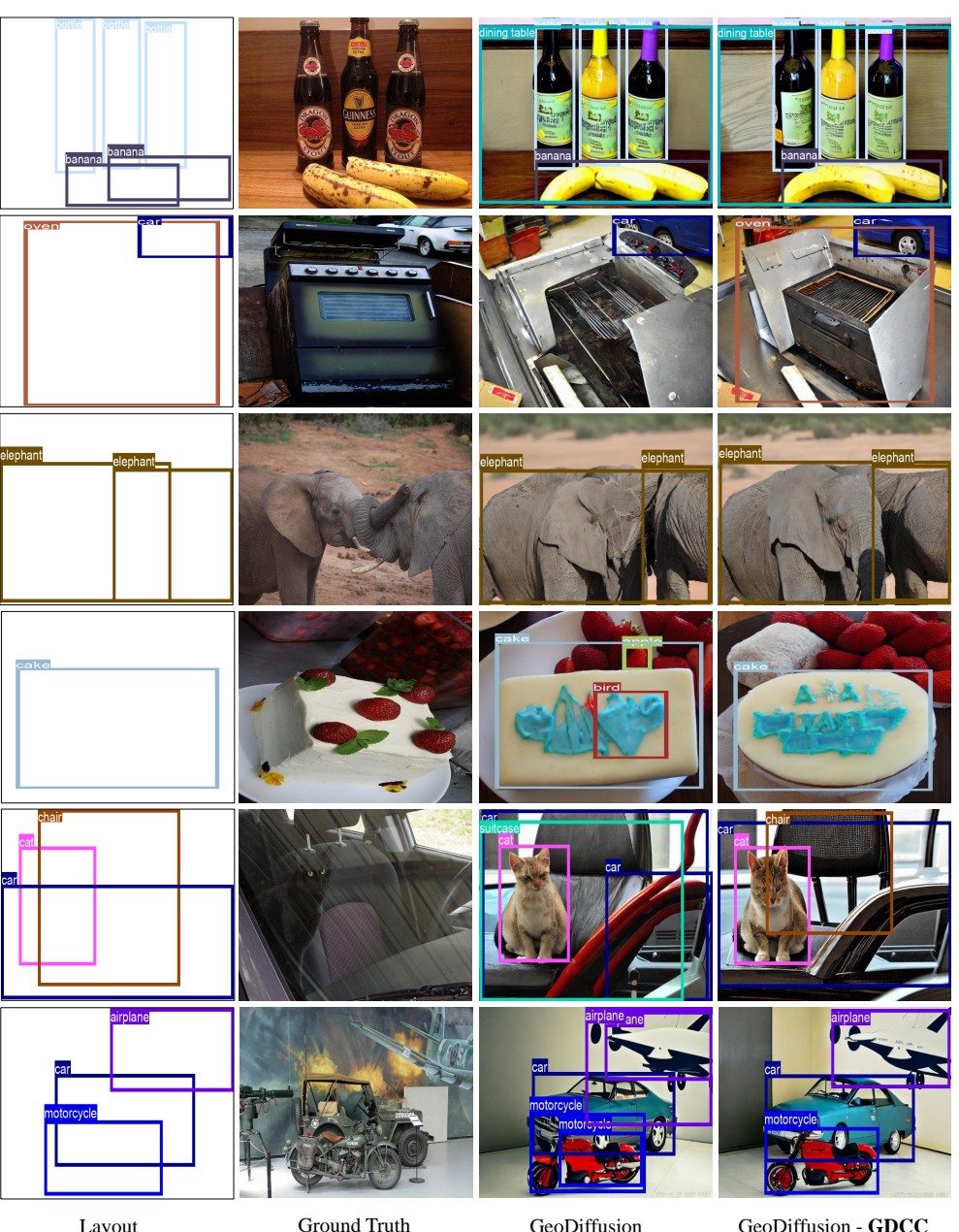

Figure 5: **More generation visual results of GeoDiffusion – GDCC on COCO 2017**. To guarantee fair comparisons, same random sampling seed is employed.

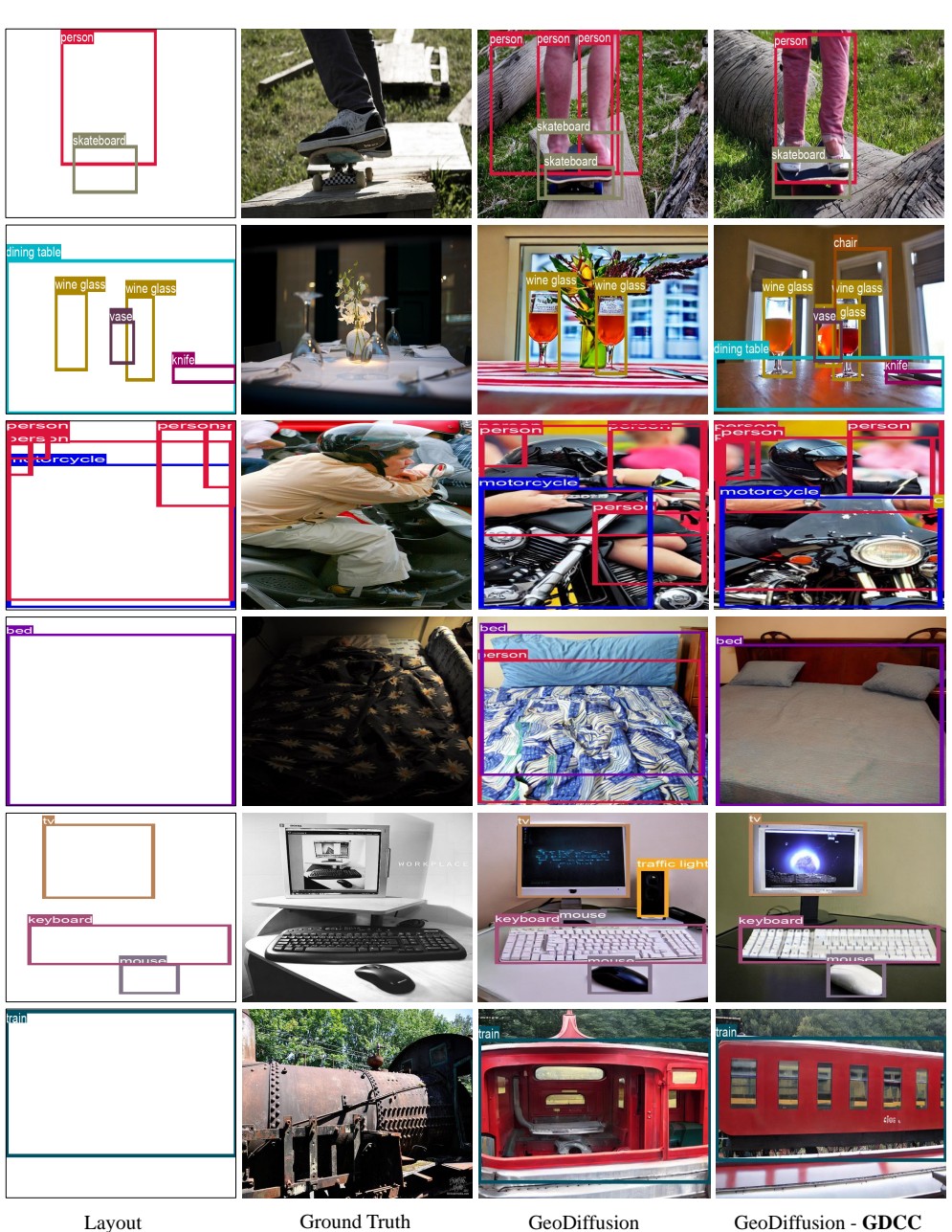

| Layout | Ground Truth | GeoDiffusion | GeoDiffusion - **GDCC** |

Figure 6: **More generation visual results of GeoDiffusion – G**DCC **on COCO 2017**. To guarantee fair comparisons, same random sampling seed is employed.

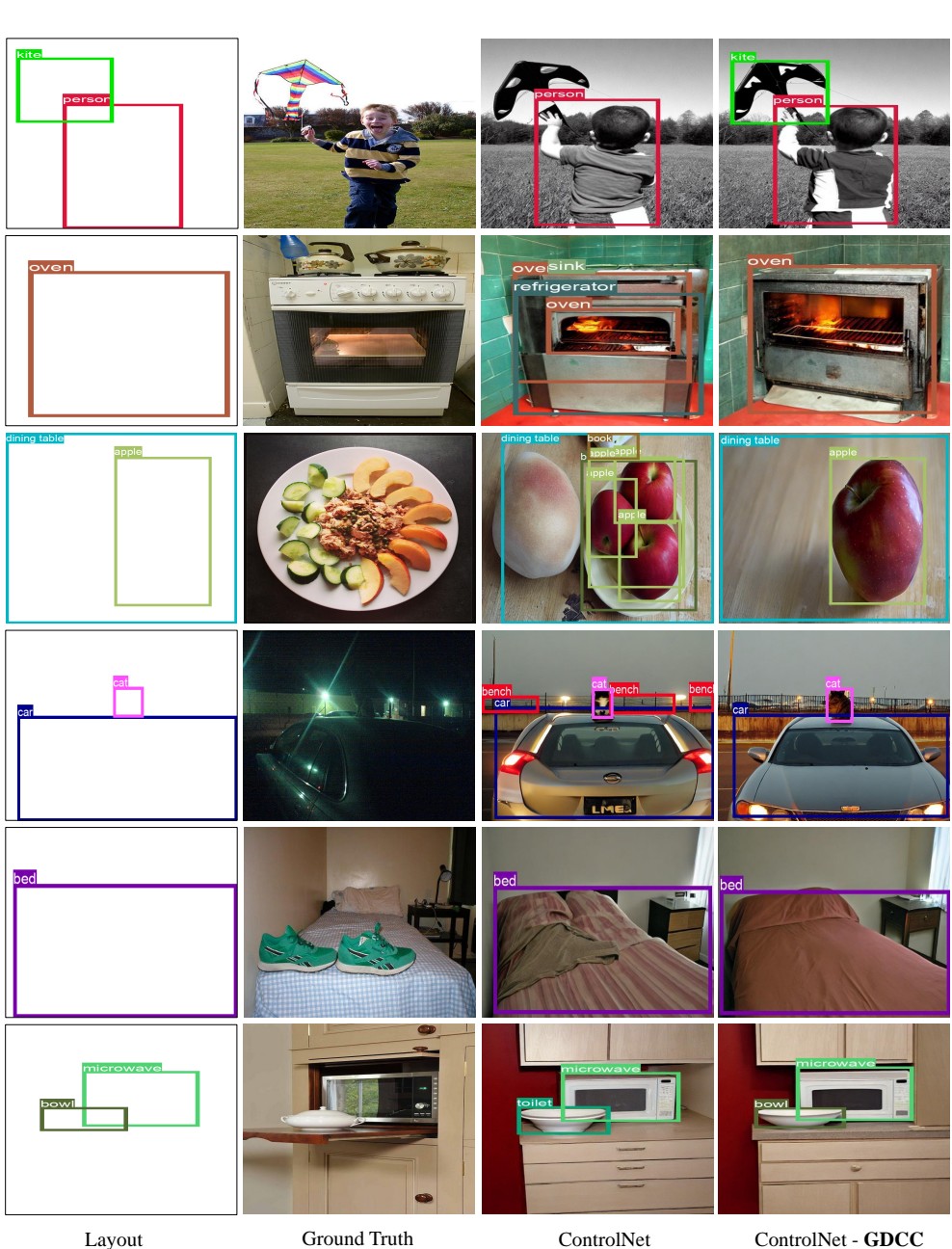

|       |              |            |                  |
|-------|--------------|------------|------------------|
| Layout | Ground Truth | ControlNet | ControlNet - **GDCC** |

Figure 7: **More generation visual results of ControlNet – G**DCC **on COCO 2017**. To guarantee fair comparisons, same random sampling seed is employed.

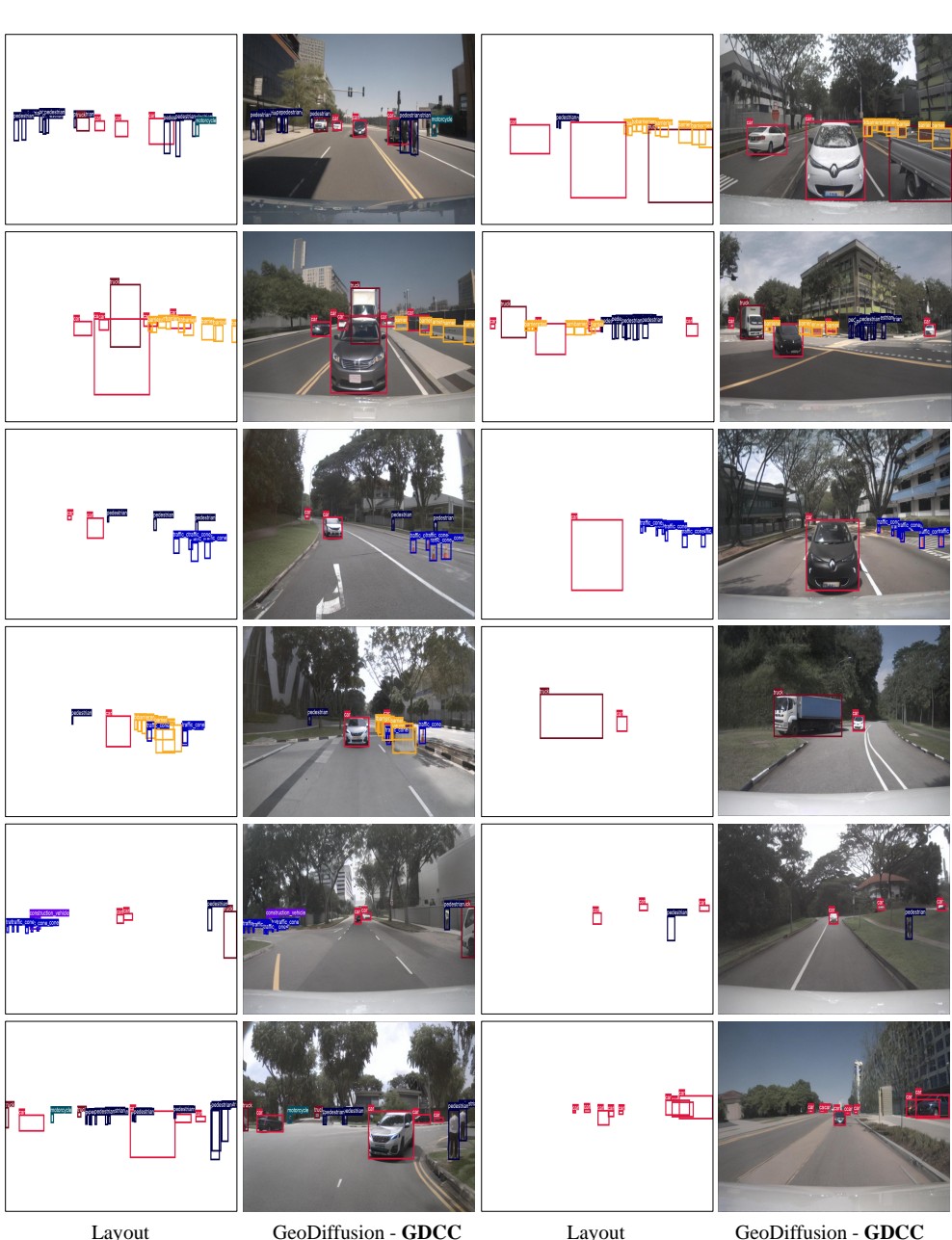

Layout      GeoDiffusion - **GDCC**      Layout      GeoDiffusion - **GDCC**

Figure 8: **More generation visual results of GeoDiffusion – G**DCC **on NuImages**. To guarantee fair comparisons, same random sampling seed is employed.

# G   MORE QUALITATIVE RESULTS OF DETECTION WITH GDCC

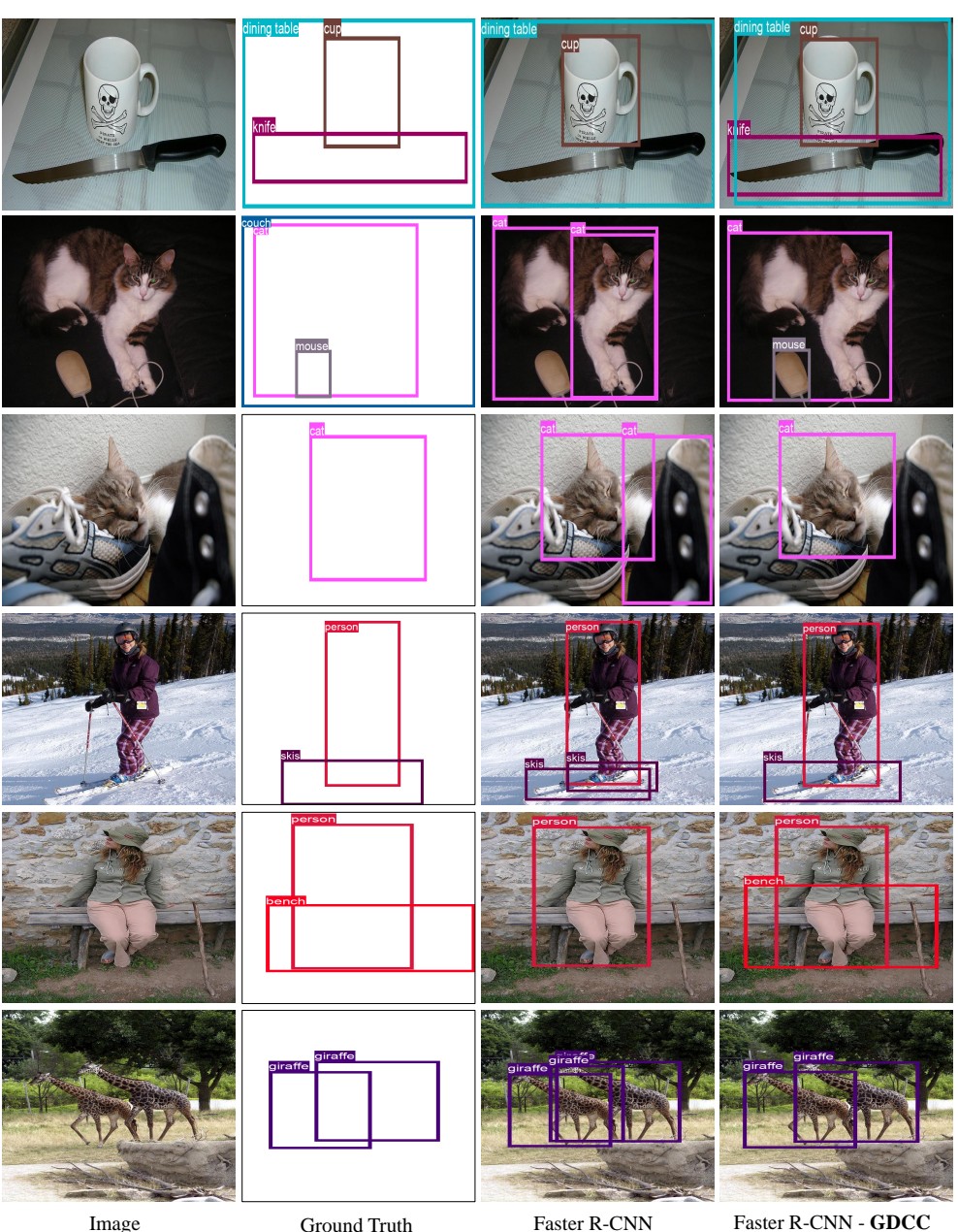

|           |              |              |                      |
|-----------|--------------|--------------|----------------------|
| Image     | Ground Truth | Faster R-CNN | Faster R-CNN - **GDCC** |

Figure 9: **More detection visual results of Faster R-CNN – GDCC on COCO 2017.**

