# OpenReview forum: "Cycle-Consistent Learning for Joint Layout-to-Image Generation and Object Detection"
_ICLR.cc/2025/Conference — ICLR 2025 Conference Withdrawn Submission_

### Official Review · Reviewer_auEv · 2024-10-24

**Soundness:** 3
**Presentation:** 3
**Contribution:** 3
**Rating:** 6
**Confidence:** 5

**Summary:**

This paper proposes GDCC, a novel framework to enhance the mutual enhancement among layout-to-image generation and object detection. Starting from a layout, GDCC first generates the corresponding image and then runs inference with an object detector, whose prediction results are further utilized to generate another image to compare with the previous image. By incorporating the perturbative single-step sampling and priority timestep re-sampling, the authors significantly improve both models while maintaining training efficiency.

**Strengths:**

- The authors provide a detailed formulation to introduce the relationship between L2I and OD.
- Not limited by the common L2I settings, the authors further study the unpaired circumstances.
- The experimental results are extensive.

**Weaknesses:**

- About the methodology:
  - Are the operations between the L2I generator and the object detector all differentiable or do you use gradient estimation methods like STE?
  - In line 244, to implement single-step sampling, the authors opt not to do Gaussian sampling as usual. So how to sample the special noise $x_t^{pert}$?
  - For unpaired circumstances in Sec. 3.2.3, since GT images are not available, I believe you need to run the whole T-step denoising for each sample in the training time, right?
  - Do you modify the layout encoding manner, or maintain the same with your baseline methods?
- About experiments:
  - Initialization: do you train from Stable Diffusion or GeoDiffusion/ControlNet? If the latter, you should clarify it more clearly in Sec. 4.1. Moreover, for the results of GeoDiffusion/ControlNet in Table 1-4, it would be better if you can fine-tune the pre-trained checkpoints also for 2/3 epochs for a fair comparison.
  - In Table 6(a), what does the variant "Baseline + $L_{ldm}$" mean, since the baseline has utilized the LDM loss?
  - In Table 6(c), do you use the same GDCC (trained with Faster-RCNN) to run experiments with different detectors?
- Overall, I think this is a solid paper, but there are several details requiring further clarification and refinement. I hope my reviews can help revise the paper.

**Questions:**

- About writing:
  - Citations: have you modified the citation format? The citations are shown in numbers in this paper, which are different from the official ICLR template.
  - Typo: In Line 345, it is easy for readers to believe GeoDiffusion is trained for 2/3 epochs.

---

### Official Review · Reviewer_nwLx · 2024-11-02

**Soundness:** 2
**Presentation:** 2
**Contribution:** 2
**Rating:** 5
**Confidence:** 3

**Summary:**

The key finding of this study is that leveraging this duality can effectively enhance the performance of both tasks. The authors introduce a Generation-Detection Cycle Consistent (GDCC) learning framework, designed to jointly optimize L2I generation and object detection (OD) in an end-to-end manner, thereby improving the performance of both models. For L2I generation, GDCC achieves up to a 2.1% improvement in FID over baseline L2I methods and shows a 2.1% increase in YOLO score, suggesting enhanced alignment between generated images and conditional layouts. For OD, GDCC achieves up to a 0.9% gain in AP, further supporting the notion of mutual enhancement between L2I generation and OD tasks.

**Strengths:**

(1): This paper is the first to identify the duality between L2I and OD tasks, leveraging this insight through the proposed Generation-Detection Cycle Consistent (GDCC) framework. The core components include a layout translation cycle loss and an image translation cycle loss.
(2): The GDCC framework jointly optimizes both detection and generation tasks in an end-to-end and Cycle-Consistent manner, enabling mutual enhancement between them.
(3): It provides a solution for handling unpaired data.

**Weaknesses:**

(1): While the authors tested the benefits of Cycle-Consistent learning for the Layout-to-Image (L2I) task using two advanced generative models, GeoDiffusion and ControlNet , they employed relatively outdated detectors such as Faster R-CNN  to evaluate the object detection task, yielding improvements of less than 1%. More experimentation with state-of-the-art detection models is needed to verify whether GDCC substantially enhances detection performance.
(2): Lack of Analysis on Iterative Cycle-Consistency: The paper does not provide an analysis of the number of iterations required for Cycle-Consistency nor address the potential risk of mode collapse over multiple iterations. Such an analysis would help clarify the stability of the iterative cycle process.
(3): Limited Evidence of Unpaired Data Benefits: The results in Table 7 indicate only a 0.3% improvement with unpaired data, which is insufficient to demonstrate its effectiveness for enhancing detection performance.

**Questions:**

(1): During training, were the generator and detector optimized in separate stages, with each component frozen when the other was being trained? Was simultaneous optimization of both components ever attempted?
(2): Why was VisorGPT chosen for handling unpaired data, and how does VisorGPT perform specifically in detection tasks?
(3): In line 309, what does LldmL_{ldm}Lldm​ refer to?
(4): For additional questions, please refer to the Weaknesses section.

---

### Official Review · Reviewer_3gfL · 2024-11-03

**Soundness:** 2
**Presentation:** 2
**Contribution:** 2
**Rating:** 5
**Confidence:** 5

**Summary:**

This paper presents a method, dubbed as GDCC (generation-detection cycle consistent), which exploits the bidirectional consistency between layout-to-image synthesis and object detection (i.e., image-to-layout). The proposed GDCC consists of two proposed loss functions, the layout translation cycle loss for improving the generator, and the image translation cycle loss for improving the object detector. The propose GDCC is tested in fine-tuning pretrained layout-to-image generators (GeoDiffusion and ControlNet) and pretrained object detector, Faster R-CNN. The generator and object detector are fine-tuned in a iterative way, fixing one and updating the other, similar in spirit to the training protocol for generative adversarial networks. To accommodate the diffusion based generators, there are modifications proposed for training efficiency and/or effectiveness, perturbative single-step sampling and priority time-step re-sampling. In experiments, the proposed method is evaluated on two datasets: COCO-Stuff train + COCO-2017 validation, and NuImages train and test splits. It obtains promising performance improvement compared with baseline generators and detectors.

**Strengths:**

+ The motivation of leveraging the duality between layout-to-image synthesis and image-to-layout detection is good.
+ The proposed two loss functions are straightforwardly designed.
+ The proposed method seems to work well in fine-tuning pretrained generators and detectors.

**Weaknesses:**

- Overall, the proposed method uses an iterative training procedure, fixing one and updating the other, which will need to address the instability issues as commonly observed in training traditional GANs,  although the proposed two motivations in the training procedure help prevent observing these issues in the experiments.
- In terms of how detectors help generators based on the proposed layout translation cycle loss (Eqn. 7),  there are missing details. Eqn.7 requires the number of predicted bounding boxes and that used in layout-to-image generation are the same, N, which can not be always guaranteed. In the appendix, the ``calculate box loss" in Algorithm 1 is not elaborated.  In experiments, small objects (<2\% of image area) are ignored, which might show up in the detection. The COCO-Stuff train dataset also contains categories that are not in the COCO pretrained Fast R-CNN. In addition, the smooth L1 loss seems not effective in helping improving the fidelity of synthesized object instance.    Overall, it brings doubts on how exactly detectors help generators.
  - The Equation indexes in Algorithms seem to be an older version, not match the main text.
- It is also not very clear how the generator helps the detector based on the image translation cycle loss (Eqn. 10). Considering the iterative nature of the proposed method, the similarity between $x_1^{syn}$ and $x_2^{syn}$ will entail less diversity of the generator.
- In the tables, some details are missing. E.g., how are the detection mAPs computed for baseline methods in comparisons.  The number of epochs is not directly comparable considering the fine-tuning nature of the proposed method.

**Questions:**

- The reviewer would like to see how authors address the first three questions in the weaknesses.
- In addition, this paper claims to be the first to identify the duality between L2I and OD. On the one hand, it is a straightforward duality. The proposed iterative optimization method seems not be an elegantly formulated solution. On the other hand, in LostGAN v2 [1], the authors explored semi-supervised settings (see Sec.2.2.6). In a follow-up work of LostGAN v2, the deep consensus learning [2], the nature of the consensus utilized in this paper has been preliminary explored.
- For the generation component, it seems to be useful to compare with some training free methods such as [3] and [4].

---
[1] LostGAN v2, https://arxiv.org/pdf/2003.11571
[2] https://arxiv.org/abs/2103.08475
[3] Training-Free Layout Control with Cross-Attention Guidance, http://arxiv.org/abs/2304.03373
[4] Grounded Text-to-Image Synthesis with Attention Refocusing, http://arxiv.org/abs/2306.05427

---

### Note · Authors · 2024-11-15

I have read and agree with the venue's withdrawal policy on behalf of myself and my co-authors.